# Positive association of angiotensin II receptor blockers, not angiotensin-converting enzyme inhibitors, with an increased vulnerability to SARS-CoV-2 infection in patients hospitalized for suspected COVID-19 pneumonia

Jean-Louis Georges[1]*, Floriane Gilles[1], Hélène Cochet[1], Alisson Bertrand[1], Marie De Tournemire[1], Victorien Monguillon[1], Maeva Pasqualini[1], Alix Prevot[1], Guillaume Roger[1], Joseph Saba[1], Joséphine Soltani[1], Mehrsa Koukabi-Fradelizi[2], Jean-Paul Beressi[3], Cécile Laureana[4], Jean-François Prost[1], Bernard Livarek[1]

1 Department of Cardiology, Centre Hospitalier de Versailles, Le Chesnay, France, 2 Emergency Department, Centre Hospitalier de Versailles, Le Chesnay, France, 3 Department of Diabetology, Centre Hospitalier de Versailles, Le Chesnay, France, 4 Department of Medical Information, Centre Hospitalier de Versailles, Le Chesnay, France

* jgeorges@ch-versailles.fr

## Abstract

### Background

Angiotensin-converting enzyme 2 is the receptor that severe acute respiratory syndrome coronavirus 2 (SARS-CoV-2) uses for entry into lung cells. Because ACE-2 may be modulated by angiotensin-converting enzyme inhibitors (ACEIs) and angiotensin II receptor blockers (ARBs), there is concern that patients treated with ACEIs and ARBs are at higher risk of coronavirus disease 2019 (COVID-19) pneumonia.

### Aim

This study sought to analyze the association of COVID-19 pneumonia with previous treatment with ACEIs and ARBs.

### Materials and methods

We retrospectively reviewed 684 consecutive patients hospitalized for suspected COVID-19 pneumonia and tested by polymerase chain reaction assay. Patients were split into two groups, according to whether (group 1, n = 484) or not (group 2, n = 250) COVID-19 was confirmed. Multivariable adjusted comparisons included a propensity score analysis.

### Results

The mean age was 63.6 ± 18.7 years, and 302 patients (44%) were female. Hypertension was present in 42.6% and 38.4% of patients in groups 1 and 2, respectively (P = 0.28). Treatment with ARBs was more frequent in group 1 than group 2 (20.7% vs. 12.0%,

**Data Availability Statement:** All relevant data are within the manuscript and its Supporting Information files.

**Funding:** The authors received no specific funding for this work.

**Competing interests:** I have read the journal's policy and the authors of this manuscript have the following competing interests: Dr. Georges has received consultant or speaker fees from AstraZeneca France, Sanofi-Aventis, Amgen, and Merck Sharpe and Dohme. The other authors have declared that no competing interests exist. This does not alter our adherence to PLOS ONE policies on sharing data and materials.

respectively; odds ratio [OR] 1.92, 95% confidence interval [CI] 1.23–2.98; P = 0.004). No difference was found for treatment with ACEIs (12.7% vs. 15.7%, respectively; OR 0.81, 95% CI 0.52–1.26; P = 0.35). Propensity score-matched multivariable logistic regression confirmed a significant association between COVID-19 and previous treatment with ARBs (adjusted OR 2.36, 95% CI 1.38–4.04; P = 0.002). Significant interaction between ARBs and ACEIs for the risk of COVID-19 was observed in patients aged > 60 years, women, and hypertensive patients.

## Conclusions

This study suggests that ACEIs and ARBs are not similarly associated with COVID-19. In this retrospective series, patients with COVID-19 pneumonia more frequently had previous treatment with ARBs compared with patients without COVID-19.

## Introduction

Coronavirus disease 2019 (COVID-19), caused by severe acute respiratory syndrome corona-virus 2 (SARS-CoV-2), was officially declared a global pandemic by the World Health Organization on 11 March 2020, and has been the greatest challenge that healthcare providers have had to face. The relationships between COVID-19 and the renin-angiotensin-aldosterone system (RAAS) and its inhibitors have been widely debated. SARS-CoV-2 uses angiotensin-converting enzyme 2 (ACE-2) as a cellular entry receptor [1,2]. ACE-2 is a key enzyme of the RAAS, which is likely to be modulated by the use of either angiotensin-converting enzyme inhibitors (ACEIs) or angiotensin II type 1 receptor blockers (ARBs) [3,4]. ACE-2 may have a protective effect against lung injury, because it degrades angiotensin (Ang) II to Ang-(1–7) [5]. The effect of RAAS inhibition on ACE-2 expression is complex [3,6,7], and has been poorly studied in humans [8,9]. In COVID-19, RAAS inhibitors could be involved on two levels: the susceptibility to SARS-CoV-2 infection; and the severity of pulmonary lesions in patients already infected.

ARBs have been demonstrated to be protective against lung injury in different experimental models of acute respiratory distress syndrome, whether infective or not [5,10–12]. ACEI/ARB treatment was associated with lower mortality in hypertensive patients already affected by COVID-19 pneumonia [13], whereas other studies failed to demonstrate a protective effect on COVID-19 severity [14].

Results of large case-control studies conducted in hypertensive patients [15] and in the general population [16–18] showed no association between ACEIs or ARBs and patients' vulnerability to COVID-19. However, in a study conducted in a large population in the USA, although the use of RAAS inhibitors was not associated with COVID-19 test positivity, hospitalizations related to COVID-19 were more frequent in patients treated with ACEIs/ARBs [17].

ACEIs and ARBs have different effects on the RAAS [3,6], as well as on the risk of non-COVID-19 pneumonia [19]; their interaction with COVID-19 may therefore differ, with the hypothesis that ACEIs could be more protective than ARBs against infection.

This study sought to compare the prevalence of hypertension and previous treatments with ACEIs and ARBs at admission in a consecutive series of high-risk patients suspected of having COVID-19 acute pneumonia, hospitalized for confirmation or not of COVID-19 in a tertiary center located in the Greater Paris area–one of the regions most affected by COVID-19 in France.

## Materials and methods

### Study design. Ethics statement

The COVHYP study is a retrospective observational study that was prospectively planned in March 2020, at the beginning of the COVID-19 outbreak in the Greater Paris area in France, and registered in May 2020 (ClinicalTrials.gov Identifier: NCT04374695). The Centre Hospitalier de Versailles is a tertiary hospital that serves a population of about 600,000 inhabitants. The study was conducted in accordance with the principles of the Declaration of Helsinki, and the protocol was approved by the French "Commission Nationale Informatique et Libertés" and a national research committee (Comité de Protection des personnes Ouest 6 –CPP 1296 HPS3; Number 2020-A01516-33). According to national regulations for non-interventional studies using medical data routinely collected from medical records, written informed consent was not mandatory. Patients and/or legal representatives received an information letter, and gave oral informed consent (non-opposition to the use of non-identifying data). Analyses were retrospective.

### Study population

From 10 March to 15 April 2020, all consecutive patients referred to the emergency department and hospitalized in a temporary 24- to 72-hour "COVID-19 screening hospitalization unit" were screened for inclusion. According to regional governmental guidelines, hospitalization was required for patients suspected of having COVID-19 who had at least one severity criterion (respiratory frequency > 22/min, spontaneous $SpO_2$ < 90%, systolic blood pressure < 90 mmHg, alteration of consciousness, fast worsening of the general status or serious dehydration in the elderly), or who had no severity criteria, but a medical history or comorbidities known to increase risk in case of COVID-19 (listed in S1 Table).

Patients were included in the study if they fulfilled the additional criteria as follows: (1) age ≥ 18 years; (2) clinical presentation suggestive of COVID-19 pneumonia (at least: fever > 38˚C or influenza-like symptoms [deep asthenia, myalgia, chills, muscular aches] associated with cough or dyspnea or need for oxygen supply [$SpO_2 \leq 90\%$]); and (3) test for the presence of SARS-CoV-2 ribonucleic acid (RNA) by reverse transcription polymerase chain reaction (RT-PCR) in nasopharyngeal or sputum samples. Exclusion criteria were the absence of clinical symptoms of COVID-19, no PCR performed, age < 18 years, prisoners or detainees, and refusal to participate.

Laboratory confirmation of SARS-CoV-2 was defined as a positive result of real-time RT-PCR assay of nasal and pharyngeal swabs, according to the French National Reference Center of Respiratory Viruses and the World Health Organization guidance [20]. As appropriate, a second RT-PCR assay from sputum or lower respiratory tract aspirates was proposed when the clinical/radiological probability of COVID-19 was high and the first RT-PCR swab assay was negative. Almost all patients underwent chest imaging by chest radiography and/or chest computed tomography (CT) scan at the emergency unit. Antihypertensive and cardiac treatments received before admission were not discontinued during the hospitalization in the COVID-19 screening hospitalization unit.

### Data collection

Clinical, radiological, and laboratory data reported in this study were collected from hospital medical reports (databases accessed from March to September 2020). The recorded data included the following: age; sex; initial symptoms; time from first symptoms suggestive of COVID-19 to admission; chest imaging performed; result of RT-PCR; serum creatinine

concentration; history of hypertension; long-term treatments for hypertension, congestive heart failure, or ischemic cardiomyopathy, including RAAS inhibitors; and medical comorbidities, such as asthma, chronic obstructive pulmonary disease, other chronic pulmonary diseases, chronic heart diseases, cancer, hypothyroidism, allergies, and immunosuppression. Chronic heart diseases included coronary artery disease (chronic coronary syndromes, history of myocardial infarction or acute coronary syndrome, history of coronary revascularization by percutaneous coronary intervention or coronary bypass graft), valvular heart diseases, hypertrophic and dilated hypokinetic cardiomyopathies, and cardiac rhythm and conduction disorders.

The estimated glomerular filtration rate (in mL/min) was calculated using the simplified Modification of Diet in Renal Disease study method [21]. Renal failure was defined by an eGRF < 60 mL/min.

Patients were considered as receiving "long-term treatment" with ACEIs, ARBs, or mineralocorticoid receptor blockers (MRBs) if they had been treated continuously within the 6 months before admission, without any switch between classes of treatments. Titration of or changes to the dose of the same ACEI/ARB treatment were accepted.

## Definition of groups

Patients were split into two groups, according to the result of the SARS-CoV-2 PCR assay, chest imaging, and clinical presentation at discharge from the "COVID-19 screening hospitalization unit". Group 1 (COVID-19) consisted of patients with a positive COVID-19 PCR assay (confirmed) and patients with symptoms and chest CT-scan abnormalities very likely to be caused by COVID-19 despite a negative PCR assay (probable). Group 2 (no COVID-19) included patients with a negative PCR assay and chest imaging not suggestive of COVID-19.

## Statistical analyses

Continuous data are presented as means ± standard deviations or medians [interquartile ranges], as appropriate, and were compared between groups using analysis of variance or the non-parametric Mann-Whitney U test. Categorical variables are presented as counts and percentages, and were compared using the $\chi^2$ test or Fischer's exact test. Multivariable analyses were performed using logistic regression, with adjustment on age, sex, obesity (body mass index > 30 kg/m$^2$), hypertension and history of chronic cardiac disease.

In addition to the main analysis, as in observational studies, treatment selection is often influenced by subject characteristics; in order to address the issues of confounding by indication, we used a propensity score-matching analysis to balance the different RAAS treatment groups on the possible baseline confounders. Multivariable logistic regressions were performed, and the probability of receiving ARBs (or ACEIs) given the observed covariates was estimated. All the variables (listed in Table 1) were included in the model, regardless of statistical significance.

After fitting the model, patients were ranked by their estimated propensity score and grouped within quintiles. Quintiles are commonly used for adjustment, as they are expected to remove 90% of the confounding. Propensity score-adjusted analyses were then performed to compare the association between COVID-19 status and previous treatments, either by univariate analyses by quintiles of propensity score in each group, or by multivariable logistic regression, including the propensity score as a covariate.

Stratified analyses were performed in prespecified subgroups, according to sex, age > 60 years, hypertension, renal failure (eGFR < 60 mL/min), and diabetes, using Cochran-Mantel-Haenszel $\chi^2$ statistics. A P-value < 0.05 was considered significant. All statistical analyses were

**Table 1. Propensity analysis: Logistic regression analysis of variables associated with a previous treatment with an ARB.**

| Analysis | Variables | P-value |
|---|---|---|
| Univariate analysis | Age | 0.000 |
| | Sex | 0.445 |
| | Hypertension | 0.000 |
| | Renal failure (eGFR < 60 mL/min) | 0.000 |
| | Diabetes | 0.003 |
| | Chronic heart disease | 0.010 |
| | Chronic respiratory disease | 0.136 |
| | Obstructive sleep apnea syndrome | 0.029 |
| | Asthma | 0.093 |
| | Obesity | 0.697 |
| Final logistic model | Age | 0.403 |
| | Sex | 0.445 |
| | Hypertension | 0.000 |
| | Renal failure (eGFR < 60 mL/min) | 0.546 |
| | Diabetes | 0.705 |
| | Chronic heart disease | 0.444 |
| | Chronic respiratory disease | 0.179 |
| | Obstructive sleep apnea syndrome | 0.200 |
| | Asthma | 0.593 |
| | Obesity | 0.641 |

ARB indicates angiotensin II receptor blocker; eGFR, estimated glomerular filtration rate.

carried out with SPSS® software, version 19.0 (SPSS Inc., Chicago, IL, USA) and R software, version i386 3.6.2.

## Results

### Baseline and initial symptoms

During the study period, 763 consecutive patients were hospitalized in the COVID-19 screening unit, 79 were excluded (S2 Table), and 684 were included in the study. COVID-19 was diagnosed in 434 patients (63.4%; 396 confirmed and 38 probable), and excluded in 250 patients (36.6%). Baseline characteristics of patients in both groups are shown in Table 2. The two groups were well balanced for fever or flu-like symptoms (almost all patients in both groups), cough (69.1% in group 1 vs. 65.6% in group 2), ear, nose and throat, digestive and neurological symptoms. Dyspnea (75.8% vs. 67.6%, respectively), male sex and time from first symptoms to admission were higher in group 1 than in group 2. A second RT-PCR sputum sample assay was performed in 55 patients (8.0%), and was positive in 17. A chest CT scan was performed most frequently in patients with subsequently confirmed COVID-19. A discrepancy between chest imaging indicated as "suggestive of COVID" by the radiologist and a discharge diagnosis of "no COVID-19" remained in seven patients, all with congestive heart failure or chronic pulmonary disease.

### Comorbidities

The distributions of comorbidities are shown in Table 3. In this series of patients, a negative association was found between COVID-19 and asthma, chronic obstructive pulmonary

**Table 2. Baseline and admission characteristics.**

|  | All patients | COVID-19 (Group 1) | No COVID-19 (Group 2) | P-value |
|---|---|---|---|---|
|  | N = 684 | N = 434 | N = 250 |  |
| Age | 63.6 ± 18.7 | 63.8 ± 17.1 | 63.2 ± 21.1 | 0.61 |
| Women | 302 (44.2) | 175 (40.3) | 127 (50.8) | < 0.01 |
| Initial symptoms |  |  |  |  |
| Fever or flu-like symptoms | 679 (99.3) | 432 (99.5) | 247 (98.8) | 0.28 |
| Cough | 464 (67.8) | 300 (69.1) | 164 (65.6) | 0.34 |
| Dyspnea | 498 (72.8) | 329 (75.8) | 169 (67.6) | 0.03 |
| Chest pain/palpitations | 115 (16.8) | 55 (12.7) | 60 (24.0) | < 0.001 |
| ENT symptoms[a] | 126 (18.4) | 84 (19.4) | 42 (16.8) | 0.41 |
| Digestive symptoms[b] | 194 (28.4) | 122 (28.1) | 72 (28.8) | 0.84 |
| Neurological symptoms[c] | 126 (18.4) | 76 (17.5) | 50 (20.0) | 0.42 |
| $SpO_2 \leq 96\%$ | 579 (84.6) | 386 (88.9) | 193 (77.2) | < 0.001 |
| Time from symptoms to admission (days) |  |  |  |  |
| Mean ± standard deviation | 6.9 ± 4.6 | 7.6 ± 4.0 | 5.7 ± 5.2 | < 0.001 |
| Median [interquartile range] | 7.0 [4.0–9.0] | 7.0 [5.0–10.0] | 4.0 [2.0–7.0] | < 0.001 |
| Admission laboratory values |  |  |  |  |
| WBC count ($10^9$/L) | 7.3 [5.3–9.7] | 6.3 [4.7–8.1] | 8.8 [8.2–12.5] | < 0.001 |
| C Reactive Protein (mg/L) | 51 [13–104] | 62 [29–124] | 19 [10–78] | < 0.001 |
| Lactate dehydrogenase (U/L) | 466 [373–629] | 554 [439–706] | 382 [365–456] | < 0.001 |
| hs Cardiac Troponin (ng/L) | 7 [4–17] | 7 [4–15] | 8 [7–19] | 0.11 |
| D-dimer (ng/mL) | 765 [370–1278] | 840 [535–1390] | 510 [375–1200] | 0.69 |
| RT-PCR for COVID-19 |  |  |  |  |
| Nasopharyngeal positive/negative | 379/305 | 379/55 | 0/250 | - |
| Sputum positive/negative | 17/38 | 17/6 | 0/32 | - |
| At least one positive PCR | 395 (57.7) | 395 (91.0) | 0 (0.0) | < 0.001 |
| Chest CT scan |  |  |  |  |
| Performed | 469 (68.8) | 320 (73.7) | 149 (59.6) | < 0.001 |
| Diagnosis of COVID-19 |  |  |  |  |
| Definite or very likely | 291 (42.5) | 284 (65.4) | 7 (2.8) | < 0.001 |
| Possible | 52 (7.6) | 24 (5.5) | 28 (11.3) |  |
| No sign of COVID-19 | 126 (18.4) | 12 (2.8) | 114 (45.6) |  |
| Extension of suspected COVID-19 lesions |  |  |  |  |
| < 10% | 77 (11.3) | 52 (12.0) | 25 (10.0) | < 0.001 |
| 10–24% | 135 (19.7) | 130 (30.0) | 5 (2.0) |  |
| 25–50% | 95 (13.9) | 94 (21.7) | 1 (0.4) |  |
| > 50% | 29 (4.2) | 29 (6.7) | 0 (0.0) |  |
| NA | 348 (50.9) | 129 (29.7) | 219 (87.6) |  |
| Admitted to intensive care unit /Need for mechanical ventilation | 66 (9.6) | 59 (13.6) | 7 (2.8) | < 0.001 |
| Hospital stay duration (days) | 8 [5–15] | 9 [5–16] | 7 [5–12]] | < 0.001 |

Data are mean ± standard deviation, number (%) or median [interquartile range]. COVID-19 indicates coronavirus disease 2019; CT, computed tomography; ENT, ear, nose, and throat; NA, not available; PCR, polymerase chain reaction; $SpO_2$, peripheral capillary oxygen saturation; WBC, white blood cell count; hs cardiac troponin, high-sensitivity cardiac troponin T test.

[a] ENT symptoms included nasal congestion, rhinorrhea, sore throat, ageusia, and anosmia.

[b] Digestive symptoms included abdominal pain, nausea, diarrhea, and poor appetite.

[c] Neurological symptoms included severe headache, severe change in behavior, convulsions, consciousness disorders, and syncope.

**Table 3. Comorbidities.**

| | All patients | COVID-19 (Group 1) | No COVID-19 (Group 2) | P-value |
|---|---|---|---|---|
| | N = 684 | N = 434 | N = 250 | |
| Asthma | 74 (10.8) | 37 (8.5) | 38 (15.2) | < 0.01 |
| Chronic pulmonary disease | 61 (8.9) | 30 (6.9) | 31 (12.4) | 0.02 |
| COPD | 50 (7.3) | 24 (5.5) | 26 (10.4) | 0.02 |
| CRPD and others | 11 (1.6) | 6 (1.4) | 5 (2.0) | 0.37 |
| Sleep apnea syndrome | 30 (4.4) | 18 (4.1) | 12 (4.8) | 0.69 |
| Diabetes mellitus | 115 (16.8) | 77 (17.6) | 38 (15.2) | 0.39 |
| Type 1 | 2 (0.3) | 2 (0.5) | 0 (0.0) | 0.40 |
| Type 2, oral treatment | 89 (13.0) | 59 (13.6) | 30 (12.0) | 0.56 |
| Type 2, insulin | 24 (3.5) | 16 (3.7) | 8 (3.2) | 0.74 |
| Obesity | 79 (11.5) | 58 (13.4) | 21 (8.4) | 0.05 |
| Hypertension | 281 (41.1) | 185 (42.6) | 96 (38.4) | 0.28 |
| Chronic heart disease | 170 (24.8) | 82 (18.9) | 64 (25.6) | 0.04 |
| Coronary artery disease | 53 (7.8) | 31 (7.1) | 22 (8.8) | 0.43 |
| Dilated cardiomyopathy | 8 (1.2) | 3 (0.7) | 5 (2.0) | 0.13 |
| Hypertrophic cardiomyopathy | 3 (0.4) | 2 (0.5) | 1 (0.4) | 0.70 |
| Valvular heart disease | 21 (3.1) | 15 (3.4) | 6 (2.4) | 0.45 |
| Arrhythmias | 85 (12.4) | 44 (10.1) | 41 (16.4) | 0.02 |
| Congestive heart failure | 22 (3.2) | 10 (2.3) | 12 (4.8) | 0.14 |
| Renal failure | | | | |
| Serum creatinine, μmol/L | 75.0 [63.0–91.0] | 76.0 [63.0–91.0] | 75.0 [62.8–92.0] | 0.31 |
| eGFR < 60 mL/min | 140/680 (20.6) | 85/432 (19.7) | 55/248 (22.2) | 0.43 |
| eGFR < 30 mL/min | 25/680 (3.7) | 12/432 (2.8) | 13/248 (5.2) | 0.10 |
| History of cancer | 106 (15.5) | 62 (14.3) | 44 (17.6) | 0.25 |
| Immunosuppression | 50 (7.3) | 28 (6.5) | 22 (8.8) | 0.26 |
| Allergies | 77 (11.3) | 51 (11.8) | 26 (10.4) | 0.59 |
| Hypothyroidism | 57 (8.3) | 43 (9.9) | 15 (6.0) | 0.08 |

Data are mean ± standard deviation, number (%) or median [interquartile range]. COPD indicates chronic obstructive pulmonary disease; COVID-19, coronavirus disease 2019; CRPD, chronic restrictive pulmonary disease; eGFR, estimated glomerular filtration rate calculated by the Modification of Diet in Renal Disease study method.

disease, and chronic heart disease. A non-significant trend towards a positive association was found for obesity and hypothyroidism. There was no difference between groups for renal function and renal failure. History of congestive heart failure or left ventricular ejection fraction < 40% was present in only 3.2% of patients (2.4% in group 1).

## Hypertension and RAAS inhibitors

Hypertension was present in 42.6% and 38.4% of patients in groups 1 and 2, respectively (P = 0.28) (Table 2), and increased with age, without differences between groups (Fig 1). Distributions of RAAS inhibitors in both groups are shown in Table 4. No patient received the combination of valsartan plus sacubitril, and one patient received both an ACEI and an ARB. The types of ACEIs and ARBs used are detailed in S3 Table. At least one RAAS inhibitor (ACEI, ARB, or MRB) was given to 34.1% of patients in group 1 and 26.8% of patients in group 2 (odds ratio [OR] 1.41, 95% confidence interval [CI] 1.00–1.99; P = 0.05). Patients in group 1 more frequently received treatment with an ARB compared with those in group 2 (20.7% vs. 12.0%, respectively; OR 1.92, 95% CI 1.23–2.98; P = 0.004). No difference was found

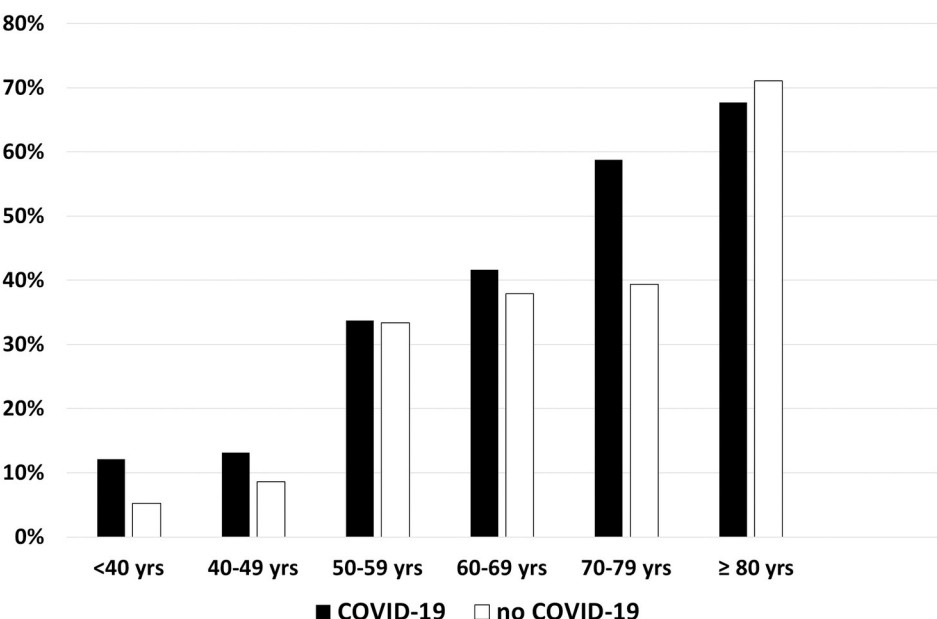

**Fig 1. Percentage of patients with hypertension by classes of age and COVID-19 status.** COVID-19 indicates coronavirus disease 2019.

**Table 4. Association between previous treatment by RAAS antagonists and COVID-19.**

| | All patients | COVID-19 (Group 1) | No COVID-19 (Group 2) | OR (95% CI) | P-value |
|---|---|---|---|---|---|
| | N = 684 | N = 434 | N = 250 | | |
| RAAS inhibitors | | | | | |
| ACEI | 93 (13.6) | 55 (12.7) | 38 (15.2) | 0.81 (0.52–1.23) | 0.35 |
| ARB | 120 (17.5) | 90 (20.7) | 30 (12.0) | 1.92 (1.23–2.98) | 0.004 |
| MRB | 6 (0.9) | 6 (1.4) | 0 (0.0) | - | 0.06 |
| ≥ 1 RAAS inhibitor[a] | 215 (31.4) | 148 (34.1) | 67 (26.8) | 1.41 (1.00–1.99) | 0.05 |
| Indication for RAAS inhibitors | | | | | |
| Hypertension | 203 (29.7) | 140 (32.2) | 63 (25.2) | 1.41 (1.00–2.00) | 0.051 |
| Congestive heart failure | 5 (0.7) | 3 (0.7) | 2 (0.8) | 0.86 (0.14–5.19) | 0.96 |
| Coronary artery disease | 21 (3.1) | 13 (3.0) | 8 (3.2) | 0.93 (0.38–2.29) | 0.90 |
| **Patients with hypertension** | **N = 281** | **N = 185** | **N = 96** | | |
| RAAS inhibitors | | | | | |
| ACEI | 83 (29.5) | 48 (25.9) | 35 (36.5) | 0.61 (0.36–1.04) | 0.07 |
| ARB | 118 (42.0) | 89 (48.1) | 29 (30.2) | 2.14 (1.28–3.59) | 0.004 |
| ≥ 1 RAAS inhibitor[a] | 203 (72.2) | 140 (75.7) | 63 (65.6) | 1.63 (0.95–2.79) | 0.08 |
| Other antihypertensive drugs[b] | 59 (21.0) | 33 (17.8) | 26 (27.1) | 0.58 (0.33–1.05) | 0.07 |
| No antihypertensive drugs | 19 (6.8) | 12 (6.5) | 7 (7.3) | 0.88 (0.34–2.32) | 0.80 |

Data are number (%) unless otherwise indicated. ACEI indicates angiotensin-converting enzyme inhibitor; ARB, angiotensin II type 1 receptor blocker; CI, confidence interval; COVID-19, coronavirus disease 2019; MRB, mineralocorticoid receptor blocker; OR, odds ratio; RAAS, renin angiotensin aldosterone system.

[a] Totals are not equal to the sums of components, due to combinations of RAAS antagonists or multiple indications for RAAS antagonists.

[b] Treatments with beta-blockers, calcium channel inhibitors or diuretics, other than RAAS antagonists.

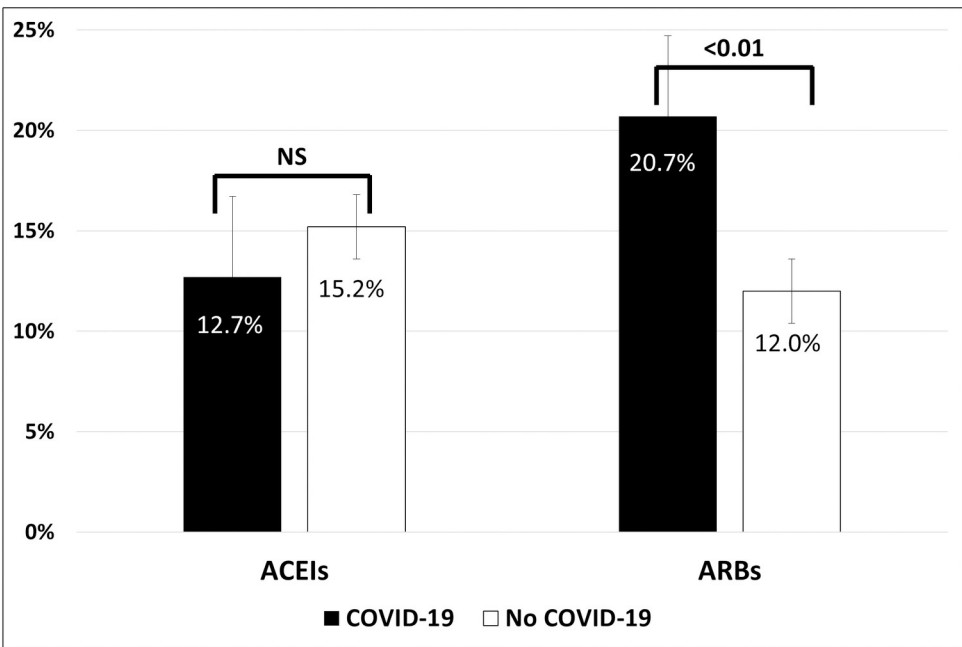

**Fig 2. Prevalence of previous treatment with ACEIs and ARBs in patients with and without COVID-19.** ACEIs indicates angiotensin-converting enzyme inhibitors; ARBs, angiotensin II type 1 receptor blockers; COVID-19, coronavirus disease 2019.

for ACEIs (12.7% vs. 15.7%, respectively; OR 0.81, 95% CI 0.52–1.26; P = 0.35) (Fig 2). Similar trends were also observed in the subgroup of hypertensive patients (Table 4).

Propensity score-matched multivariable logistic regression confirmed a significant association between COVID-19 and previous treatment with ARBs (adjusted OR 2.36, 95% CI 1.38–4.04; P = 0.002) (Tables 1, 5 and 6).

Similar results were found in two additional analyses where patients with "probable COVID-19" were excluded from group 1 (comparison of 396 patients with PCR-confirmed COVID-19 and 250 patients without COVID-19) or attributed to group 2 (patients without COVID-19) (S4 and S5 Tables).

**Table 5. Propensity analysis: Association between previous treatment with ARBs and COVID-19 pneumonia according to the quintiles of propensity score.**

| Quintile | | COVID-19 | No COVID-19 | P-value |
|---|---|---|---|---|
| Q1 | N | 81 | 55 | |
| | ARB, n (%) | 0 (0.0) | 1 (1.8) | 0.41 |
| Q2 | N | 91 | 45 | |
| | ARB, n (%) | 1 (1.1) | 0 (0.0) | 0.67 |
| Q3 | N | 80 | 56 | |
| | ARB, n (%) | 0 (0.0) | 0 (0.0) | - |
| Q4 | N | 99 | 37 | |
| | ARB, n (%) | 42 (42.4) | 13 (35.1) | 0.29 |
| Q5 | N | 81 | 55 | |
| | ARB, n (%) | 47 (58.0) | 16 (29.1) | 0.001 |

ARB indicates angiotensin II type 1 receptor blocker; COVID-19, coronavirus disease 2019.

**Table 6. Propensity analysis: Logistic regression analysis of previous treatment with ARB and COVID-19 pneumonia adjusted on propensity score.**

|  | B | E.S. | Wald | dfl | P-value | OR (95% CI) |
|---|---|---|---|---|---|---|
| Age | 0.004 | 0.005 | 0.429 | 1 | 0.512 | 1.004 (0.993–1.014) |
| Sex | −0.477 | 0.173 | 7.548 | 1 | 0.006 | 0.621 (0.442–0.872) |
| Hypertension | 1.443 | 1.081 | 1.781 | 1 | 0.182 | 4.232 (0.508–35.219) |
| eGFR < 60 mL/min | −0.188 | 0.239 | 0.617 | 1 | 0.432 | 0.828 (0.518–1.325) |
| Diabetes | −0.005 | 0.242 | 0.000 | 1 | 0.984 | 0.995 (0.619–1.600) |
| Chronic cardiac disease | −0.225 | 0.271 | 0.686 | 1 | 0.407 | 0.799 (0.469–1.359) |
| Chronic pulmonary disease | −0.615 | 0.321 | 3.671 | 1 | 0.055 | 0.541 (0.288–1.014) |
| Sleep apnea syndrome | −0.391 | 0.470 | 0.691 | 1 | 0.406 | 0.676 (0.269–1.700) |
| Asthma | −0.488 | 0.263 | 3.432 | 1 | 0.064 | 0.614 (0.366–1.029) |
| Obesity | 0.547 | 0.300 | 3.312 | 1 | 0.069 | 1.728 (0.959–3.114) |
| Propensity score for ARB | −3.850 | 2.622 | 2.157 | 1 | 0.142 | 0.021 (0.000–3.625) |
| ARB (yes/no) | 0.857 | 0.275 | 9.709 | 1 | 0.002 | 2.357 (1.375–4.042) |
| Constant | 0.640 | 0.332 | 3.719 | 1 | 0.054 | 1.897 |

ARB indicates angiotensin II type 1 receptor blocker; CI, confidence interval; COVID-19, coronavirus disease 2019; eGFR, estimated glomerular filtration rate calculated by the Modification of Diet in Renal Disease study method; OR, odds ratio.

## Subgroup analyses

Stratified analyses (Fig 3) showed opposite ORs for the risk of COVID-19 associated with previous ARBs and ACEIs in women, patients aged > 60 years, and hypertensive patients. In these groups, the risk of COVID-19 was significantly increased in patients receiving ARBs, and significantly (borderline for hypertension) reduced in patients treated with ACEIs, the P-value for interaction being significant. A less contrasted similar pattern, without significant interaction, was found for diabetes and renal failure.

## Discussion

The results of this study, conducted on a consecutive series of patients hospitalized with a clinical presentation consistent with COVID-19 pneumonia, showed a positive association between COVID-19 and previous treatment with ARBs, and no association with ACEIs. Opposite risk ratios for COVID-19, protective for ACEIs and not protective for ARBs, were found in patients aged > 60 years and women, with a significant interaction. These results suggest that long-term treatment with a RAAS inhibitor may be not neutral for vulnerability to SARS-CoV-2.

There are theoretical arguments for different effects of ACEIs and ARBs on the RAAS, and vulnerability to pulmonary infection. Both ACEIs and ARBs have been shown to increase cardiac ACE-2 gene transcription in some animal models [4,5], but there is no evidence that RAAS inhibitors upregulate transmembrane ACE-2 receptor expression in the human lung [22]. Moreover, several experimental and clinical data suggest that ACEIs and ARBs do not have similar effects on ACE-2 expression and activity. In a murine model of myocardial ischemia, the upregulation of ACE-2 induced by lisinopril was higher than that induced by losartan, but was associated with no increase in cardiac ACE-2 activity. In the same model, lisinopril and losartan were associated with opposite variations in plasma Ang II and the Ang(1–7)/Ang II ratio [3]. Conflicting evidence was also reported with ramipril, which failed to increase ACE-2 [8]. Discrepant effects of ACEIs and ARBs on ACE-2 mRNA and activity, as well as on RAAS metabolism, have been summarized by Kreutz et al. [6].

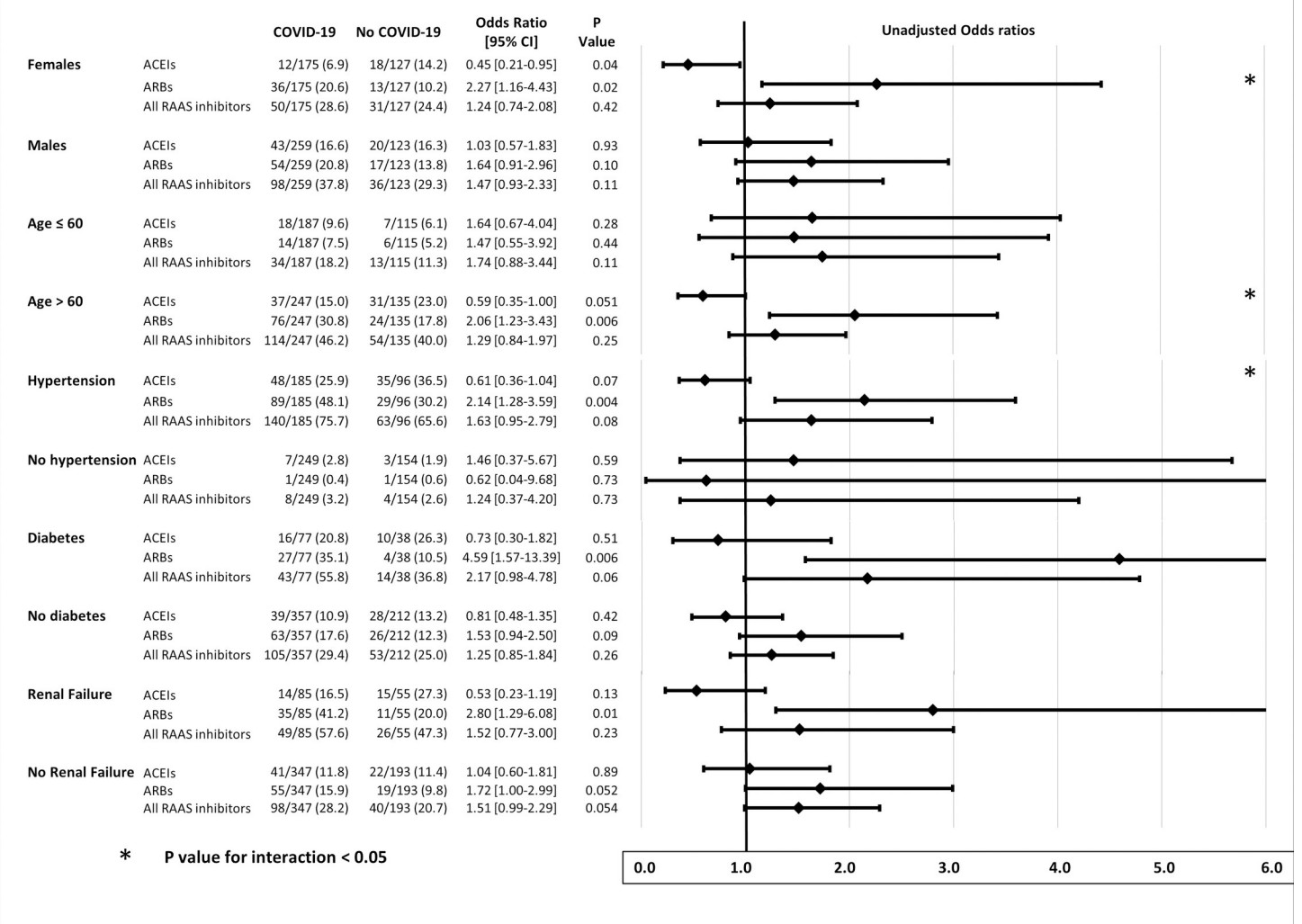

**Fig 3. Stratified analysis of relationships between previous treatment with RAAS blockers and COVID-19, according to sex, age > 60 years, hypertension, diabetes, and renal failure (eGFR < 60 mL/min).** ACEIs indicates angiotensin-converting enzyme inhibitors; ARBs, angiotensin II type 1 receptor blockers; COVID-19: coronavirus disease 2019; eGFR, estimated glomerular filtration rate calculated by the Modification of Diet in Renal Disease study method; RAAS, renin-angiotensin-aldosterone system.

Our results are partly discrepant with other observational retrospective studies conducted in the USA [17,18], Italy [16], and Denmark [15], which found no global difference in the prevalence of RAAS inhibitors between COVID-19 patients and controls.

However, these studies strongly differ from the present study by the selection of either patients or controls. Two studies [17,18] compared consecutive patients tested for COVID-19, regardless of hospitalization. For two other studies, age- and sex-matched controls were drawn from general population databases, and were not specifically tested for COVID-19 [15,16]. The prevalence of treatment with RAAS inhibitors varied considerably in the overall study populations, from 12.5% and 18.4% in the studies in the USA [17,18] to 45.6% in the Italian study [16], and > 60% in the Danish study, which was restricted to hypertensive patients [15].

In all these studies, baseline characteristics and comorbidities were different in cases and controls. In the present study, both patients and controls were patients who presented to the emergency hospital department with symptoms suggestive of acute pulmonary infection, and

who were admitted because of severity criteria, including the need for oxygen supply. The more selective inclusion criteria resulted in baseline characteristics, clinical symptoms, and comorbidities being relatively well balanced between groups, despite the absence of randomization, with the exception that COVID-19 patients had a more severe respiratory presentation (more dyspnea, lower SpO$_2$, higher extension of pulmonary lesions on CT scan, more admissions to the intensive care unit) than controls. These differences in selection criteria may explain, in part, the difference in the results. Interestingly, in one study [17], a significant association was found between ACEI/ARB treatment and hospitalization, with an OR (1.93, 95% CI 1.38–2.71) close to that found for ARBs in our analysis. Another study found a positive association between RAAS blockers and the risk of COVID-19, which was explained by a higher prevalence of cardiovascular disease [16].

A key finding of our study was that the association between ARB treatment and the risk of COVID-19 remained significant when taking into account major confounding factors. Moreover, we identified subgroups of patients for whom opposite effects of ARBs and ACEIs on the risk of COVID-19 were found. In women and patients aged > 60 years, and in a lesser extent in patients with hypertension, diabetes, and moderate renal failure (eGRF < 60 mL/min), the risk of COVID-19 was twice as high in patients treated with ARBs compared with those not treated with ARBs, whereas previous treatment with ACEIs appeared protective, with ORs for COVID-19 significantly < 1 in women and borderline non-significant in hypertensive patients and those aged > 60 years (Fig 3). Gender differences in relationships between ACEI/ARBs and vulnerability to COVID-19 may be important, as it has been shown that women with hypertension are less frequently treated with ACEIs and ARBs than men [23].

Therefore, although RAAS inhibitors do not appear to be associated with COVID-19 in the general population [15–18], our study suggests that, among a specific subset of patients with significant comorbidities and a more severe clinical presentation, ARBs have a negative effect, whereas ACEIs do not. These results have to be confirmed. Until results of confirmatory studies are available, and because discontinuation of ARBs may be harmful in high-risk patients [24], recommendations to continue RAAS inhibitors in patients affected by or at high risk of COVID-19 should be respected [25].

## Study limitations

This study has limitations. Although it was prospectively designed, collection and analyses of data were retrospective. The biases classically associated with retrospective studies may account for the observed differences. Particularly, misclassification of patients with and without COVID-19 may have occurred. In the study 38/384 of patients from group 1 were diagnosed as having probable COVID-19 despite a negative PCR assay. However, this false negative rate of 10% compares favorably with that of 30% reported in Wuhan, China [26]. Conversely, few patients who had a negative PCR assay were classified as "no COVID-19", although abnormalities in the chest CT scan were consistent with COVID-19. In order to take into account and overcome this putative bias, additional analyses were done, excluding the 38 patients with probable COVID-19 (S3 Table), and then pooling the probable COVID-19 with the non-COVID-19 patients (S4 Table). Similar results were found in the first case, and borderline non-significant results in the second case (the least favorable to the hypothesis of a significant association between ARBs and COVID-19). Analyses were adjusted on propensity scores, taking into account variables that were independently associated with previous treatment with ARBs or ACEIs. Propensity score-adjusted analyses confirmed the positive association between ARBs and COVID-19. Finally, data on treatment with non-steroid anti-inflammatory agents were not collected and no adjustment was made on this.

## Conclusions

This study confirmed that, overall, RAAS blockers are not associated with the risk of COVID-19. However, comparative analyses suggested that ACEIs and ARBs are not similarly associated with COVID-19 incidence, as patients with COVID-19 pneumonia had been treated previously with ARBs more frequently than patients without COVID-19. An opposite effect of ACEIs, likely to be protective, and ARBs, not protective, was observed in women, patients aged > 60, and, to a lesser extent, hypertensive patients. The results of the present study need to be interpreted with caution, given the retrospective monocentric observational design of the study. These results have to be confirmed, and do not question the current recommendations to continue long-term treatment with ACEIs and ARBs, particularly in patients already infected by SARS-CoV-2.

## Supporting information

**S1 Table. High Council of Public Health, France: Definition of population with a risk of developing severe COVID-19 (March 31, 2020).**
(DOC)

**S2 Table. COVHYP study: Causes for exclusion.**
(DOC)

**S3 Table. ACEIs and ARBs used in the study.**
(DOC)

**S4 Table. Association between PCR-confirmed COVID-19 and long-term treatment with RAAS antagonists: The medium hypothesis (the 38 "probable" COVID-19 patients are excluded from analysis, which compares patients with PCR-confirmed COVID-19 and patients without COVID-19).**
(DOC)

**S5 Table. Association between results of PCR for COVID-19 and long-term treatment with RAAS antagonists: The worst hypothesis (all "probable" COVID-19 patients are classified as no-COVID-19 patients).**
(DOC)

## Acknowledgments

We thank Stéphanie Marque Juillet, MD, Catherine Palette, Pharm.D, and all the biologists and technicians of the department of virology, Maxime de Malherbe, MD, François Mignon, MD, Pénélope Labauge, MD, and all the radiologists and radiology technicians of the radiology department, and the physicians and nurses of the emergency department, and the departments of diabetology and cardiology of the Centre Hospitalier de Versailles. We thank Jean-Baptiste Azowa and Karelle Aumasson for their technical assistance, and help in the data collection. We thank Pr Michel Azizi, MD, PhD, Sophie Rushton-Smith, PhD, and the Centre Hospitalier de Versailles for editorial assistance.

## Author Contributions

**Conceptualization:** Jean-Louis Georges, Mehrsa Koukabi-Fradelizi, Jean-Paul Beressi, Jean-François Prost.

**Data curation:** Hélène Cochet, Alisson Bertrand, Marie De Tournemire, Victorien Monguillon, Maeva Pasqualini, Alix Prevot, Guillaume Roger, Joseph Saba, Joséphine Soltani.

**Formal analysis:** Jean-Louis Georges, Cécile Laureana.

**Investigation:** Floriane Gilles.

**Methodology:** Jean-Louis Georges, Guillaume Roger, Jean-Paul Beressi, Jean-François Prost.

**Project administration:** Jean-Louis Georges, Mehrsa Koukabi-Fradelizi, Jean-Paul Beressi, Bernard Livarek.

**Resources:** Cécile Laureana.

**Software:** Hélène Cochet, Cécile Laureana.

**Supervision:** Jean-François Prost, Bernard Livarek.

**Validation:** Floriane Gilles, Hélène Cochet, Alisson Bertrand, Marie De Tournemire, Victorien Monguillon, Maeva Pasqualini, Alix Prevot, Guillaume Roger, Joseph Saba, Joséphine Soltani, Mehrsa Koukabi-Fradelizi, Jean-Paul Beressi, Jean-François Prost, Bernard Livarek.

**Visualization:** Hélène Cochet, Jean-François Prost, Bernard Livarek.

**Writing – original draft:** Guillaume Roger.

**Writing – review & editing:** Jean-Louis Georges.

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
