## [Decision Letter · Decision Letter 0]

26 Aug 2020

PONE-D-20-22693

Positive association of Angiotensin II Receptor Blockers, not Angiotensin-Converting Enzyme Inhibitors, with an increased vulnerability to SARS-CoV-2 infection in patients hospitalized for suspected COVID-19 pneumonia

PLOS ONE

Dear Dr. Georges,

Thank you for submitting your manuscript to PLOS ONE. After careful consideration, we feel that it has merit but does not fully meet PLOS ONE’s publication criteria as it currently stands. Therefore, we invite you to submit a revised version of the manuscript that addresses the points raised during the review process.

Please see comments by the reviewers. Kindly submit point by point response in your revised version of the manuscript.

We look forward to receiving your revised manuscript.

Kind regards,

Muhammad Adrish

Academic Editor

PLOS ONE

Journal Requirements:

2. Please include the date(s) on which you accessed the databases or records to obtain the data used in your study.

"I have read the journal's policy and the authors of this manuscript have the following competing interests: Dr. Georges has received consultant or speaker fees from AstraZeneca France, Sanofi-Aventis, Amgen, and Merck Sharpe and Dohme. The other authors have declared that no competing interests exist."

Reviewers' comments:

Reviewer's Responses to Questions

**Comments to the Author**

1. Is the manuscript technically sound, and do the data support the conclusions?

Reviewer #1: Yes

Reviewer #2: Yes

Reviewer #3: Partly

2. Has the statistical analysis been performed appropriately and rigorously? 

Reviewer #1: I Don't Know

Reviewer #2: Yes

Reviewer #3: No

3. Have the authors made all data underlying the findings in their manuscript fully available?

Reviewer #1: Yes

Reviewer #2: Yes

Reviewer #3: No

4. Is the manuscript presented in an intelligible fashion and written in standard English?

Reviewer #1: Yes

Reviewer #2: Yes

Reviewer #3: Yes

5. Review Comments to the Author

Reviewer #1: The study investigated the susceptibility of the patients who were treated with angiotensin converting enzyme inhibitors or angiotensin II receptor blockers. The authors suggested that significant association between COVID-19 pneumonia and history of taking ARBs. Although this study has remarkable contents, some questions are raised.

Reviewer #2: Georges and coworkers presented an interesting study: Positive association of ARB, not ACEI, with an increased vulnerability to SARS-CoV-2 infection in patients hospitalized for suspected COVID-19 pneumonia.

The authors enrolled almost 684 patients hospitalized patients, used very elegant methodology, presented sub-analysis of the enrolled population of patients, stressed the limitations of their study, found some interesting results and presented some interesting conclusions.

1. Authors should define more precisely the “long-term” treatment for hypertension (years, months, some previous switching of therapy between ARBs and ACEI?).

2. Data on the use of NSAIDs and prevalence of chronic kidney disease are missing, data would be useful as both entities could influence the prescribing of RAAS blockade drugs.

The authors should accept and discuss these comments in the manuscript.

Reviewer #3: Manuscript ID: PONE-D-20-22693 Thank you for giving us the opportunity to review the manuscript Title: “Positive association of Angiotensin II Receptor Blockers, not Angiotensin-Converting Enzyme Inhibitors, with an increased vulnerability to SARS-CoV-2 infection in patients hospitalized for suspected COVID-19 pneumonia” A manuscript in which the author described comparative analysis suggested that ACEIs and ARBs are not similarly associated with the COVID-19 incidence, the patients with COVID-19 pneumonia receiving more frequently a previous treatment with ARBs than patients without COVID-19. We have some points we would like to refer:

Major Comments:

- Lack of data in the study regarding types of ACEI and ARBS .

- No available data of previously patient compliance of antihypertensive drugs or not.

- Statistical analysis should be reviewed for values as missing P values for variables and subgroup analysis result of each variable as diabetes and extension of suspected COVID-19 etc.

- More detailed study definitions are needed.

- Grammatical and alphabetical adjustment is recommened.

6. PLOS authors have the option to publish the peer review history of their article (what does this mean?). If published, this will include your full peer review and any attached files.

Reviewer #1: No

Reviewer #2: No

Reviewer #3: No

---

## [Author Response · Author response to Decision Letter 0]

8 Oct 2020

Reviewer #1: 

General comments:

The study investigated the susceptibility of the patients who were treated with angiotensin converting enzyme inhibitors or angiotensin II receptor blockers. The authors suggested that significant association between COVID-19 pneumonia and history of taking ARBs. Although this study has remarkable contents, some questions are raised.

Major comments:

1. What do the authors think about the reason of discrepancy between the result of this study and previous reports regarding effects of ACEI and ARB with COVID-19 pneumonia?

Answer: The four previous studies with negative results strongly differ from the present study by the selection of either patients or controls. Two studies compared consecutive patients tested for COVID-19 regardless of hospitalization. For two other studies, age- and sex-matched controls were drawn from general population databases, and were not specifically tested for COVID-19.

The present study included patients with intermediate severity criteria, having symptoms of acute pneumonia, and requiring hospitalization, whatever their positive or negative status for COVID-19. This may explain why baseline characteristics of cases and controls were not so different, and why adjusted and propensity-matched analyses did not change the case-control comparisons in the present study, in contrast to some previous studies. This may also account for different effects of RAAS inhibitors in this specific study population.

2. Please describe the contents of the discussion section more concisely and conclusively.

Answer: (shared answer to comments 2 and 3) The Discussion section has been shortened and extensively modified in order to comply with the reviewer’s comments 1, 2, and 3 (pages 14–17, lines 276–363).

We have tried to point out the new academic content of the study, and emphasized the results of stratified analyses that have not been performed previously. We hope that the discussion has been improved by these changes, as suggested by the reviewer.

3. As the authors have already described in the discussion section, there are similar studies already published with similar and opposite conclusions. It would be better if authors point out new academic contents of the present study in the discussion session.

 Answer: (shared answer to comments 2 and 3).

Minor comments:

1. The authors should define which diseases were included in chronic heart disease. 

Answer: “Chronic heart disease” included coronary artery disease (chronic coronary syndromes, history of myocardial infarction or acute coronary syndrome, history of coronary revascularization by PCI or CABG), valvular heart diseases, hypertrophic and dilated hypokinetic cardiomyopathies, and cardiac rhythm and conduction disorders. The four main categories of chronic heart disease are detailed in Table 3. We added a sentence to the “Materials and methods” section (page 5, lines 125–129) (see also answer to comment from reviewer #3 “More detailed study definitions are needed”). 

2. In the text, there are two different terms were used and it can cause confusion to the readers. 

1) SARSCoV-2, SARS-Cov-2, SARS-CoV-2 and SARS-COV-2

2) Angiotensin-converting enzyme type 2 and angiotensin-converting enzyme 2

3) RAAS system and RAAS

Answer: The terms have been homogenized: SARS-CoV-2; angiotensin-converting enzyme 2; and RAAS as abbreviation for renin-angiotensin-aldosterone system.

3. The authors used abbreviation before explanation of full term.

1) RNA

2) RT-PCR

3) RAAS

4) ENT

Answer: This has been corrected. Thank you.

Reviewer #2: Georges and coworkers presented an interesting study: Positive association of ARB, not ACEI, with an increased vulnerability to SARS-CoV-2 infection in patients hospitalized for suspected COVID-19 pneumonia.

The authors enrolled almost 684 patients hospitalized patients, used very elegant methodology, presented sub-analysis of the enrolled population of patients, stressed the limitations of their study, found some interesting results and presented some interesting conclusions.

1. Authors should define more precisely the “long-term” treatment for hypertension (years, months, some previous switching of therapy between ARBs and ACEI?).

Answer: In this study, “long-term” treatment for hypertension was defined as continuous oral therapy by ACEI or ARB for at least 6 months, without interruption. No case of switch between ARB and ACEI within the 6 months before inclusion was reported by the patients, including the few patients with congestive heart failure. This detail has been added to the “Materials and methods” section (page 5, lines 133–136).

2. Data on the use of NSAIDs and prevalence of chronic kidney disease are missing, data would be useful as both entities could influence the prescribing of RAAS blockade drugs.

Answer: In order to comply with the reviewer’s comment, we have now collected the serum creatinine concentration for all patients but four (680/684), allowing us to take this important factor into account in the analyses. The eGFR was calculated with the MDRD method, and the definition of renal failure (eGRF < 60 mL/min) has been added to the “Materials and methods” section (page 5, lines 121 and 130–132). All analyses were performed again, controlling on renal function. Stratified analyses were also done in patients with eGRF < 60 mL/min and ≥ 60 mL/min. Results, Tables, and Fig 3 were modified in order to take into account data on renal function. 

No data could be collected on the use of NSAIDs, and this was added to the limitations of the study (Page 17, lines 351–352).

The authors should accept and discuss these comments in the manuscript.

Reviewer #3: Manuscript ID: PONE-D-20-22693 Thank you for giving us the opportunity to review the manuscript Title: “Positive association of Angiotensin II Receptor Blockers, not Angiotensin-Converting Enzyme Inhibitors, with an increased vulnerability to SARS-CoV-2 infection in patients hospitalized for suspected COVID-19 pneumonia” A manuscript in which the author described comparative analysis suggested that ACEIs and ARBs are not similarly associated with the COVID-19 incidence, the patients with COVID-19 pneumonia receiving more frequently a previous treatment with ARBs than patients without COVID-19. We have some points we would like to refer:

Major comments:

1. Lack of data in the study regarding types of ACEI and ARBS .

Answer: the different ACEIs and ARBs are now detailed in a supplementary table (S3 Table). A sentence has been added to the “Results” section (page 11, lines 217–218). The two main ACEIs were ramipril and perindopril, and the two main ARBs were candesartan and irbesartan, with no differences between groups. 

2. No available data of previously patient compliance of antihypertensive drugs or not.

Answer: This point is important. At the inclusion visit, we try to collect the maximal information available on the medical treatment within the 6 months before admission. Patients and/or their relatives, as appropriate, were asked about the current treatment, with a focus on antihypertensive and cardiological treatments, and the start date (or month and year) of each medication. We also asked whether any discontinuation of the ACEI/ARB treatment, or a switch between ACEI and ARB occurred over the 6 previous months. Patients were considered as treated with an ACEI or ARB only if they had received an ACEI or an ARB continuously without any switch. Titration of or changes to the dose of the same ACEI/ARB treatment were accepted. No other specific data on compliance were obtained.

3. Statistical analysis should be reviewed for values as missing P values for variables and subgroup analysis result of each variable as diabetes and extension of suspected COVID-19 etc.

Answer: We thank the reviewer for this comment. All statistical analyses have been redone, including and controlling on renal function, as suggested by reviewer 2. The missing P values in the stratified analysis were added to Tables 2 and 3, and Fig 3, and odds ratios have been added to Table 4. Remaining missing P values are voluntary (Table 2), because formal comparisons were not appropriate for some items. 

4. More detailed study definitions are needed.

Answer: as suggested by the reviewer, and the other reviewers (see reviewer #1, minor 1 and reviewer #2, comment 1), some definitions have been added

Definition of “chronic heart disease” (page 5, lines 125–129 and Table 3).

Definition of “long-term” treatment by RAAS inhibitors: continuous treatment by the same class of treatment (ACEI or ARB), for ≥ 6 months before admission, no switch between the two classes (page 5, lines 133–136).

Definition of renal failure using the MDRD estimate of glomerular filtration rate (page 5, lines 130–132) (one reference has been added).

5. Grammatical and alphabetical adjustment is recommended.

Answer: as suggested by the reviewer, the text has been revised for the English language, by a native English-speaking external editor

---

## [Decision Letter · Decision Letter 1]

2 Nov 2020

PONE-D-20-22693R1

Positive association of angiotensin II receptor blockers, not angiotensin-converting enzyme inhibitors, with an increased vulnerability to SARS-CoV-2 infection in patients hospitalized for suspected COVID-19 pneumonia

PLOS ONE

Dear Dr. Georges,

Thank you for submitting your manuscript to PLOS ONE. After careful consideration, we feel that it has merit but does not fully meet PLOS ONE’s publication criteria as it currently stands. Therefore, we invite you to submit a revised version of the manuscript that addresses the points raised during the review process.

ACADEMIC EDITOR: Please see comments made by the reviewers and provide point by point response in your revised manuscript

We look forward to receiving your revised manuscript.

Kind regards,

Muhammad Adrish

Academic Editor

PLOS ONE

Reviewers' comments:

Reviewer's Responses to Questions

**Comments to the Author**

1. If the authors have adequately addressed your comments raised in a previous round of review and you feel that this manuscript is now acceptable for publication, you may indicate that here to bypass the “Comments to the Author” section, enter your conflict of interest statement in the “Confidential to Editor” section, and submit your "Accept" recommendation.

Reviewer #1: All comments have been addressed

Reviewer #2: All comments have been addressed

Reviewer #3: All comments have been addressed

2. Is the manuscript technically sound, and do the data support the conclusions?

Reviewer #1: Yes

Reviewer #2: Yes

Reviewer #3: Yes

3. Has the statistical analysis been performed appropriately and rigorously? 

Reviewer #1: Yes

Reviewer #2: Yes

Reviewer #3: N/A

4. Have the authors made all data underlying the findings in their manuscript fully available?

Reviewer #1: Yes

Reviewer #2: Yes

Reviewer #3: Yes

5. Is the manuscript presented in an intelligible fashion and written in standard English?

Reviewer #1: Yes

Reviewer #2: Yes

Reviewer #3: Yes

6. Review Comments to the Author

Reviewer #1: The authors revised manuscript appropriately. The quality of this manuscript has improved. I suggest accept without further revision.

Reviewer #2: Georges and coworkers presented an interesting study: Positive association of ARB, not ACEI, with an increased vulnerability to SARS-CoV-2 infection in patients hospitalized for suspected COVID-19 pneumonia.

The authors correctly accepted suggested comments.

Reviewer #3: Manuscript ID: PONE-D-20-22693R1 Thank you for giving us the opportunity to review the manuscript Title: “Positive association of Angiotensin II Receptor Blockers, not Angiotensin-Converting Enzyme Inhibitors, with an increased vulnerability to SARS-CoV-2 infection in patients hospitalized for suspected COVID-19 pneumonia” A manuscript in which the author described comparative analysis suggested that ACEIs and ARBs are not similarly associated with the COVID-19 incidence, the patients with COVID-19 pneumonia receiving more frequently a previous treatment with ARBs than patients without COVID-19. We have some points we would like to refer:

Comments:

- Please add the laboratory value in baseline characteristics either correlated to covid-19 infection as CRP serum ferritin,

and LDH or correlated to thromboembolic phase of the disease highly sensitive troponin and D-dimer.

- Prognostic factors like hospital stay, need for mechanical ventilation between two groups

- Table 6 what does K means?

7. PLOS authors have the option to publish the peer review history of their article (what does this mean?). If published, this will include your full peer review and any attached files.

Reviewer #1: No

Reviewer #2: **Yes: **Sebastjan Bevc

Reviewer #3: No

---

## [Author Response · Author response to Decision Letter 1]

13 Nov 2020

Reviewer #1: The authors revised manuscript appropriately. The quality of this manuscript has improved. I suggest accept without further revision.

Answer: We thank you for your comments and the time dedicated to our manuscript. 

Reviewer #2: Georges and coworkers presented an interesting study: Positive association of ARB, not ACEI, with an increased vulnerability to SARS-CoV-2 infection in patients hospitalized for suspected COVID-19 pneumonia.

The authors correctly accepted suggested comments.

Answer: We thank you for your comments and the time dedicated to our manuscript.

Reviewer #3: Manuscript ID: PONE-D-20-22693R1 Thank you for giving us the opportunity to review the manuscript Title: “Positive association of Angiotensin II Receptor Blockers, not Angiotensin-Converting Enzyme Inhibitors, with an increased vulnerability to SARS-CoV-2 infection in patients hospitalized for suspected COVID-19 pneumonia” A manuscript in which the author described comparative analysis suggested that ACEIs and ARBs are not similarly associated with the COVID-19 incidence, the patients with COVID-19 pneumonia receiving more frequently a previous treatment with ARBs than patients without COVID-19. We have some points we would like to refer:

Comments:

- Please add the laboratory value in baseline characteristics either correlated to covid-19 infection as CRP serum ferritin, and LDH or correlated to thromboembolic phase of the disease highly sensitive troponin and D-dimer.

Answer: Results of WBC, CRP, LDH, hs cardiac troponin T, and D dimer have been added in baseline characteristics (Table 1, pages 8-9), as suggested. Serum ferritin has not been added; ferritin was not measured in routine and was available for <10 patients only. 

- Prognostic factors like hospital stay, need for mechanical ventilation between two groups

Answer: All study patients with or without COVID-19 were hospitalized because they required oxygen supply. All patients admitted in ICU received a mechanical ventilation, and this was added in the Table 1. Hospital stay duration was also added in the Table 1.

- Table 6 what does K means? 

Answer: K indicates the constant in the regression model. In the revised version, we replaced K by “Constant”, more explicit.

---

## [Decision Letter · Decision Letter 2]

9 Dec 2020

Positive association of angiotensin II receptor blockers, not angiotensin-converting enzyme inhibitors, with an increased vulnerability to SARS-CoV-2 infection in patients hospitalized for suspected COVID-19 pneumonia

PONE-D-20-22693R2

Dear Dr. Georges,

We’re pleased to inform you that your manuscript has been judged scientifically suitable for publication and will be formally accepted for publication once it meets all outstanding technical requirements.

Kind regards,

Muhammad Adrish

Academic Editor

PLOS ONE

Additional Editor Comments (optional):

You have satisfactorily answered all queries.

Reviewers' comments:

Reviewer's Responses to Questions

**Comments to the Author**

1. If the authors have adequately addressed your comments raised in a previous round of review and you feel that this manuscript is now acceptable for publication, you may indicate that here to bypass the “Comments to the Author” section, enter your conflict of interest statement in the “Confidential to Editor” section, and submit your "Accept" recommendation.

Reviewer #3: All comments have been addressed

2. Is the manuscript technically sound, and do the data support the conclusions?

Reviewer #3: Yes

3. Has the statistical analysis been performed appropriately and rigorously? 

Reviewer #3: Yes

4. Have the authors made all data underlying the findings in their manuscript fully available?

Reviewer #3: Yes

5. Is the manuscript presented in an intelligible fashion and written in standard English?

Reviewer #3: Yes

6. Review Comments to the Author

Reviewer #3: Manuscript ID: PONE-D-20-22693R2 Thank you for giving us the opportunity to review the manuscript Title: “Positive association of Angiotensin II Receptor Blockers, not Angiotensin-Converting Enzyme Inhibitors, with an increased vulnerability to SARS-CoV-2 infection in patients hospitalized for suspected COVID-19 pneumonia” A manuscript in which the author described comparative analysis suggested that ACEIs and ARBs are not similarly associated with the COVID-19 incidence, the patients with COVID-19 pneumonia receiving more frequently a previous treatment with ARBs than patients without COVID-19.

I accept this revised version.

7. PLOS authors have the option to publish the peer review history of their article (what does this mean?). If published, this will include your full peer review and any attached files.

Reviewer #3: No

---

## [Editor Report · Acceptance letter]

11 Dec 2020

PONE-D-20-22693R2 

Positive association of angiotensin II receptor blockers, not angiotensin-converting enzyme inhibitors, with an increased vulnerability to SARS-CoV-2 infection in patients hospitalized for suspected COVID-19 pneumonia 

Dear Dr. Georges:

I'm pleased to inform you that your manuscript has been deemed suitable for publication in PLOS ONE. Congratulations! Your manuscript is now with our production department. 

Kind regards, 

on behalf of

Dr. Muhammad Adrish 

Academic Editor

PLOS ONE